

# Estimation of aerosol complex refractive indices for both fine and coarse modes simultaneously based on AERONET remote sensing products

Ying Zhang[1], Zhengqiang Li[1], Yuhuan Zhang[2], Donghui Li[1], Lili Qie[1], Huizheng Che[3], and Hua Xu[1]

[1]State Environmental Protection Key Laboratory of Satellite Remote Sensing, Institute of Remote Sensing and Digital Earth, Chinese Academy of Sciences, Beijing 100101, China

[2]Satellite Environment Center, Ministry of Environmental Protection, Beijing 100094, China

[3]Chinese Academy of Meteorological Sciences, Chinese Meteorological Administration, Beijing 100081, China

*Correspondence to:* Zhengqiang Li (lizq@radi.ac.cn)

**Abstract.** Climate change assessment, especially model evaluation, needs to know complex refractive indices (CRI) of atmospheric aerosols, separately for both fine and coarse modes. However, the widely used aerosol CRI obtained by the global Aerosol Robotic Network (AERONET), correspond to total-column aerosol particles without separation for fine and coarse modes. This paper establishes a method to separate CRIs of fine and coarse particles based on AERONET aerosol products, including volume particle size distribution (VPSD), aerosol optical depth (AOD) and absorbing AOD. The method consists of VPSD breaking-down and sub-mode CRI separation parts and yields spectral CRIs for both fine and coarse modes simultaneously. Numerical experiment shows good performance for typical water-soluble, biomass burning and dust aerosol types and the estimated uncertainties on the retrieved sub-mode CRIs are about 0.11 (real part) and 78% (imaginary part), respectively. One year measurements at AERONET Beijing site are processed and we obtain CRIs of 1.48-0.010i (imaginary part at 440 nm is 0.012) for fine mode particles and 1.49-0.004i (imaginary part at 440 nm is 0.007) for coarse mode particles, for the period of 2014-2015. Our results also suggest that both aerosol fine and coarse mode CRIs have distinct seasonal characteristics, particularly CRIs of fine particles in winter season are significantly higher than summer, due to possible anthropogenic influences.

## 1 Introduction

Complex Refractive Indices (CRI) of aerosols, describing scattering and absorption properties of atmospheric particulate matters, are important parameters affecting calculation of short-wave radiative budget and aerosol climate effect. Improving the knowledge on aerosol CRI is of great interests to decrease the uncertainties associated to aerosols in the climate change assessment (Boucher et al., 2013). Since 20th century, direct measurement approaches of CRI of small particles have been developed in the laboratory (e.g. Woo et al, 2013; Mogo et al., 2012; Marley et al., 2001; Patterson et al., 1977; Volz, 1973). As to aerosol particles in the real atmosphere, many studies retrieved CRI of near ground surface aerosols (e.g. Kostenidou et al., 2007; Malloy et al., 2009; Dinar et al., 2006; McMurry et al., 2002), through in situ measurements of particle size distribution



as well as scattering and absorption coefficients. Meanwhile, several remote sensing methods were developed to obtain CRI of total-column atmospheric aerosols (e.g. Raut and Chazette, 2007; Li et al., 2006; Sinyuk et al., 2003; Dubovik and King, 2000; Kaufman et al., 2001, Wendisch and von Hoyningen-Huene, 1994; Nakajima et al., 1983). As Nakajima et al. (1983) and Wendisch and von Hoyningen-Huene (1994) reported, the aerosol CRIs can be retrieved by using spectral aerosol optical depth

and diffusely scattered radiances. One of widely recognized CRI remote sensing approach is the statistically optimal estimation method based on Sun/sky-radiometer measurements (Dubovik and King, 2000), which has been successfully implemented in the world-wide Aerosol Robotic Network (AERONET) (Holben et al., 1998, 2001). In addition, Li et al. (2006) supplemented the polarized sky radiance measurements to better constrain AERONET CRI retrievals. Raut and Chazette (2007) also synthesized the Lidar measurements to obtain CRI of aerosols confined within planetary boundary layer.

Although above-mentioned remote sensing methods retrieve CRI of total column aerosols, it still remains a big challenge to obtain CRI simultaneously for different modes (e.g. fine and coarse modes, respectively). CRI of fine and coarse modes may differ significantly, due to different compositions and sources (Marley et al., 2001). For example, fine modes are mainly determined by anthropogenic emission or nucleation process, while coarse modes are dominated by natural sources of wind-blown dust or sea salt (Willeke and Whitby, 1975). As to atmospheric models, e.g. the global three-dimensional chemical transport

model (GEOS-Chem) and the Community Multi-scale Air Quality model (CMAQ), aerosols radiative properties are simulated based on source emission (i.e. fine and coarse sources separately) inventories, and thus knowledge on CRI of different aerosol modes are essential to validate model performance for the assessment of aerosol climate effects. There are only few studies (e.g. Xu et al., 2015; Wu et al., 2015) attempted to retrieve CRI of both fine and coarse modes simultaneously from advanced ground-based remote sensing measurements, e.g. multi-spectral polarized sky radiance. Meanwhile, most of AERONET sites

provide official CRI products without distinguishing fine and coarse modes. Considering the essential values of world-wide, long-term continuous and high quality AERONET CRI dataset, it is valuable to develop an approach to separate CRI for both fine and coarse modes, based on directly AERONET official aerosol products, instead of developing an entire algorithm performing retrieval from radiance level.

In this paper, we introduce a method to separate CRI of both fine and coarse modes from AERONET aerosol products (Sec-

tion 2). Section 3 presents the theoretical simulation and analyses and Section 4 focus on the results of 1-yr measurements in Beijing. The results are summarized in Section 5.

## 2  Method

The ground-based Sun/sky-radiometer is one of major instrument observing total column atmospheric aerosol properties. There are several long-term Sun/sky-radiometer networks operated regionally or globally, e.g. AERONET (Holben et al., 1998, 2001)

and SKYNET (Hashimoto et al., 2012). Several inversion algorithms (e.g. King et al., 1978; Nakajima et al, 1996; Dubovik and King, 2000, 2006; Li et al., 2006) have been developed based on Sun/sky-radiometer to retrieve aerosol parameters, like Aerosol Optical Depth ($\tau$), Single-Scattering Albedo, Absorbing Aerosol Optical Depth ($\tau_a$), Volume Particle Size Distribution (VPSD), real ($n(\lambda)$) and imaginary ($k(\lambda)$) parts of CRI corresponding to total-column atmospheric aerosols. Although VPSD





are retrieved for a wide radius range (e.g. 0.05-15 $\mu m$) with multi-bins (e.g. 22) showing fine and coarse modes clearly, the $n$ and $k$ parts of CRI are still commonly assumed constant for both fine and coarse modes. In practice, separation of CRI for both fine and coarse modes depends on precisely breaking-down VPSD into individual modes and dealing with spectral variation of CRI. Prior to these key steps, a framework to characterize AERONET aerosol products with both fine and coarse mode is

needed to establish.

## 2.1 Aerosol characterization framework based on AERONET products

In order to separate CRI for different modes, we need to characterize AERONET aerosol products by two major assumptions: (i) AERONET VPSD can be fitted by multi-peak Log-Normal Modes (LNM). We choose multi-modal log-normal distributions to fit the AERONET retrieved VPSD by the following formula:

$$\frac{dV(r)}{dlnr} = \sum_{i=1}^{m} \frac{C_i}{\sqrt{2\pi}|ln\sigma_i|} exp\left[-\frac{1}{2}\left(\frac{lnr-lnr_i}{ln\sigma_i}\right)^2\right] \quad m = 1, 2, ... \tag{1}$$

where $dV/dlnr$ (in unit of $\mu m^3/\mu m^2$) is the volume particle size distribution, $C_i$ ($\mu m^3/\mu m^2$) and $r_i$ ($\mu m$) and $\sigma_i$ are the volume modal concentration, median radius and standard deviation of each LNM mode, respectively. For most of cases, AERONET VPSD can be separated by two LNMs (i.e. $m = 2$ in Eq. (1)) with fine and coarse modes corresponding to small size and large size peak LNM, respectively. When $m$ is larger than 2, all peaks with radius $r_i$ less than 1.0 $\mu m$ can be considered

as belonging to the fine mode, and others belonging to the coarse mode. (ii) Fine and coarse modes have their own sub-CRIs, while real part ($n$) of sub-CRIs is spectrally independent, and imaginary part ($k$) of sub-CRIs have spectral variation follows:

$$n_{f/c}(\lambda) = n_{f/c} \quad \lambda = 440, 675, 870, 1020nm \tag{2}$$

$$k_{f/c}(\lambda) = \begin{cases} k_{f/c,440} & \lambda = 440nm \\ k_{f/c} & \lambda = 675, 870, 1020nm \end{cases} \tag{3}$$

where $\lambda$ denotes standard wavelength of AERONET products, $f$ and $c$ represent fine and coarse modes, respectively.

The above assumed spectral properties of sub-CRIs are useful to simplify subsequence procedure and it basically fits current knowledge on aerosol properties. Fig.1 shows CRI of various aerosol components, including black carbon, dust, organics, sulfate and aerosol water. Based on these data, CRI real parts ($n$) of aerosol components are quite constant from UV to near infrared spectral region. Only Hematite, following Sokolik and Toon (1999), shows some spectral variation, but its content

is usually very low in aerosols, e.g. less than 5% in mass (Schuster et al., 2015; Wagner et al., 2012; Lafon et al., 2004). In contrast, CRI imaginary parts ($k$) of aerosol components show significant spectral variation, especially at short wavelengths (e.g. 440 nm). Again, Hematite shows strongly higher absorption at 440 nm so that it can affect $k$ value of entire aerosols although with low concentration. In addition, the imaginary part of organics shows some spectral variation (e.g. a difference of 0.123 between 350 and 500 nm) while that of black carbon mixture also has a smaller spectral change, e.g. variation of about

0.05 at short wavelength.





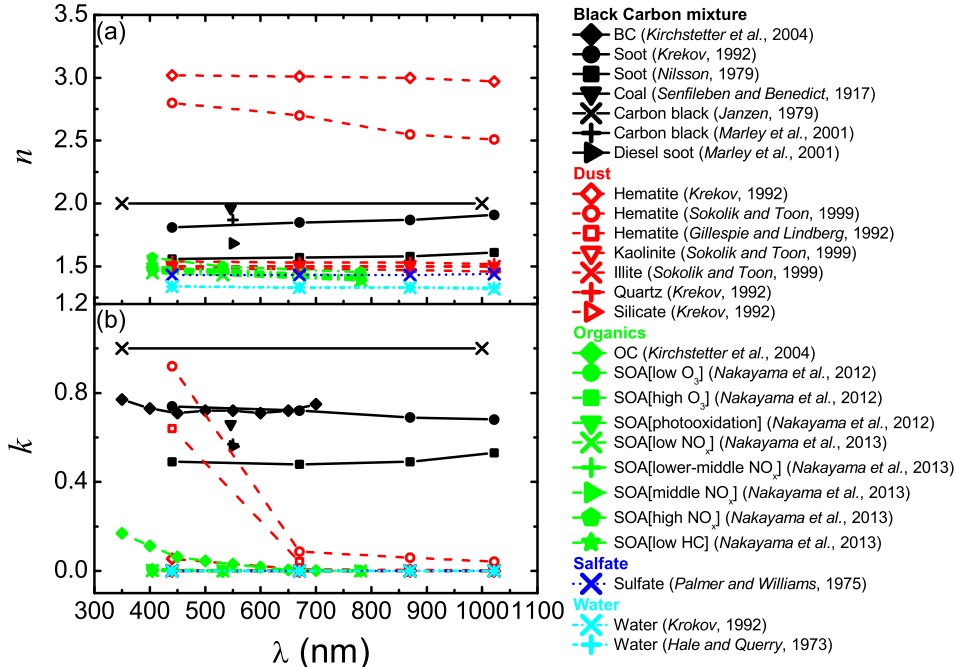

**Figure 1.** Complex refractive indices (real part: a; imaginary part: b) reported in literatures. Abbr. BC (Black Carbon), OC (Organic Carbon), SOA (Secondary Organic Carbon).

## 2.2 Size distribution breaking-down

Optical parameters (e.g. $\tau$ and $\tau_a$) of aerosols are usually sensitive to the size distribution and less sensitive to the refractive indices (Zhang et al., 2013; Gobbi et al., 2007). Therefore, the separation of aerosol VPSD is the basis for the next steps of accurate estimation of sub-CRIs. The traditional size distribution breaking-down approaches, e.g. cutting off fine and coarse modes from VPSD based on a fixed or dynamic particle radius, cannot naturally separate fine and coarse modes, especially the obtained sub-mode curve is not a complete log-normal function. Therefore, in this study, we separate the VPSD into complete log-normal functions following the VPSD breaking-down method described in Cuesta et al., (2008). Firstly, we need to set the initial guess values of $r_i$, $\sigma_i$ and $C_i$ in Eq. (1). This is based on calculation of the second derivative of VPSD and follows the Eq.4:

$$
\begin{cases}
g\left(r\right) = -\dfrac{d^2\nu(r)}{dr^2} \\
r_i = g^{-1}\left(max_i\left(g\left(r\right)\right)\right) \quad i = 1, m \\
\sigma_i = \sqrt{\dfrac{r_i^+}{r_i^-}} \\
C_i = \nu\left(r_i\right)
\end{cases}
\tag{4}
$$

where, $\nu$ (instead of $dV/dlnr$) is AERONET VPSD, and $r_i^+$ and $r_i^-$ are the zero-crossing points of $g(r)$ around $r_i$ in each mode. Then, to obtain the optimized values of these three parameters (i.e. $C_i$, $r_i$ and $\sigma_i$) of each peak, an iterative procedure is





performed by minimizing Chi-Square on VPSD (see Eq.5) using the NelderMead simplex algorithm (Lagarias et al., 1998).

$$\chi^2 = \sum_{j=1}^{22} \frac{\left(\nu\left(r_j\right) - \nu_{calc}\left(r_j\right)\right)^2}{\nu\left(r_j\right)} \tag{5}$$

where, $\nu$ and $\nu_{calc}$ are $dV/dlnr$ from AERONET products and re-calculated from the separated fine and coarse-mode parameters, respectively. And $j$ is the bins of AERONET VPSD.

## 2.3 Separating refractive indices for fine and coarse modes

According to the aerosol characterization framework in Section 2.1, the flowchart (Fig. 2) of the fine and coarse mode CRI separation is as follows (based on the separated size distribution in Section 2.2):

a). Guesses of 6 output parameters ($n_f$, $k_{f,440}$, $k_c$; $n_c$, $k_{c,440}$, $k_c$). The initial guess values are set with AERONET product values: $n_f = n_{AERONET}(440 \text{ nm})$, $k_{f,440}$ and $k_f = k_{AERONET}(440 \text{ nm})$, $n_c = n_{AERONET}(870 \text{ nm})$, $k_{c,440}$ and $k_c = k_{AERONET}(870 \text{ nm})$. Meanwhile, the boundary ranges of these parameters are set as: $n_f$ [1.33, 1.6], $n_c$ [1.33, 1.6], $k_{f,440}$ [0.0, 0.5], $k_{c,440}$ [0.0, 0.5], $k_f$ [0.0001, 0.5] and $k_c$ [0.0001, 0.5].

b). Calculating effective CRI corresponding to each VPSD bins. Here, based on the guessed CRIs of both fine and coarse modes in the previous step, we employ an internal mixing approach, following volume average rule (Heller, 1965), to estimate CRI of each particle radius bins:

$$n\left(r\right) = \frac{n_f \nu_f(r) + n_c \nu_c(r)}{\nu_f(r) + \nu_c(r)}$$
$$k\left(\lambda, r\right) = \frac{k_f \nu_f(r) + k_c \nu_c(r)}{\nu_f(r) + \nu_c(r)} \tag{6}$$

These CRI of each bins can be used to improve the precision of calculation of aerosol optical parameters, employed by next constrain steps.

c). Calculating aerosol optical parameters ($\tau$ for $\lambda$=440, 500, 675, 870, 1020 nm and $\tau_a$ for $\lambda$=440, 675, 870, 1020 nm, here wavelengths correspond to AERONET product bands) with the use of the aerosol CRI and VPSD of step b), by Mie theory:

$$\tau\left(\lambda\right) = \int \pi r^2 Q_{ex}\left(\lambda, r, n\left(r\right) - ik\left(\lambda, r\right)\right) \cdot \frac{dN(r)}{dr} \cdot dr$$
$$\tau_s\left(\lambda\right) = \int \pi r^2 Q_{sc}\left(\lambda, r, n\left(r\right) - ik\left(\lambda, r\right)\right) \cdot \frac{dN(r)}{dr} \cdot dr \tag{7}$$
$$\tau_a\left(\lambda\right) = \tau\left(\lambda\right) - \tau_s\left(\lambda\right)$$

where $dN/dr$ represents the number size distribution in the atmospheric column which can be pbtained from VPSD, $\tau_s$ is the scattering aerosol optical depth, $\lambda$ is wavelength and $r$ is particle radius. $Q_{ex}$ and $Q_{sc}$, represent the extinction and scattering efficiency of single spherical particles, respectively.

d). Calculation of Jacobians of $\tau(\lambda)$ and $\tau_a(\lambda)$, by disturbing each sub-CRI parameters by 0.1% ($\Delta$), which is needed by the optimization algorithm in step e).

e). Find the optimal solution based on an Limited-memory optimization algorithm (BFGS: Broyden–Fletcher–Goldfarb–Shanno) (Zhu et al., 1997) by constraining both $\tau(\lambda)$ and $\tau_a(\lambda)$ with AERONET products.

f). Check if the convergence, $(f_i - f_{i+1})/max(f_{i+1}, f_i, 1) < \eta \cdot \epsilon$, achieves. If yes, output the separated sub-CRI parameters,



otherwise replace the initial guess with current solution and repeat steps b)-f). Here, $f$ is the Chi-square kernel function, with subscript $i$ and $i+1$ representing iteration counts, $\eta$ is a convergence control factor and $\epsilon$ is machine precision (typically setting $\eta = 10^{-4}/\epsilon$).

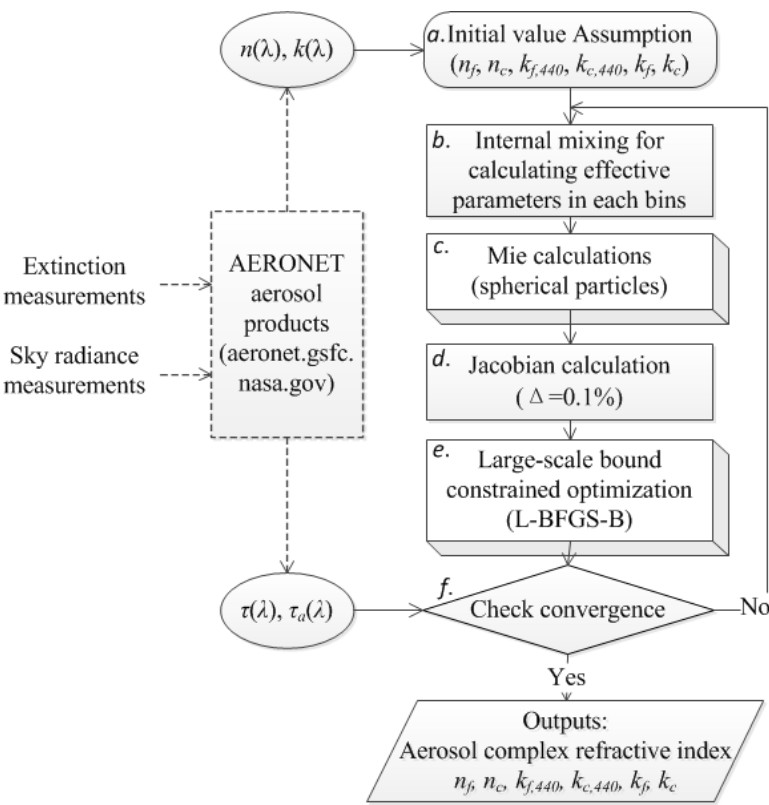

**Figure 2.** The flowchart of fine and coarse modes CRI estimation scheme.

## 3 Numerical tests

### 3.1 Typical aerosol model test

To test the CRI separation scheme, we employ three typical (i.e. water-soluble, biomass burning and dust) aerosol type models (Table 1) in this paper. These aerosol type parameters are the same with Dubovik et al. (2000), except for supplementing coarse mode $n_c$ and $k_c$ (1.53-0.008i) for all types and keeping original CRIs for only fine modes. Based on these microphysical characters, $\tau$ and $\tau_a$ (table 1) and diffused sky radiance are calculated by Mie and radiative transfer code. Then, these sky radiance are inverted by AERONET inversion algorithm (Dubovik et al., 2000) to yield AERONET aerosol CRI products ($n$ and $k$), which will be used as the inputs of our estimation scheme.





**Table 1.** Typical aerosol models (WS: Water-soluble, BB: Biomass burning, DU: Dust) and their sub-mode parameters.

| Type | | | | | | | | | | AOD & AAOD by Mie | | Simulated AERONET products by radiance inversion | |
|---|---|---|---|---|---|---|---|---|---|---|---|---|---|
| | | | | Fine and coarse mode parameters | | | | | | | | | |
| | $r_1$ | $r_2$ | $\sigma_1$ | $\sigma_2$ | $C_1/C_2$ | $n_f$ | $k_f$ | $n_c$ | $k_c$ | $\tau$ (440/500/675/ 870/1020 nm) | $\tau_a$ (440/675/ 870/1020 nm) | $n$ (440/675/ 870/1020 nm) | $k$ (440/675/ 870/1020 nm) |
| WS | 0.118 | 1.17 | 0.6 | 0.6 | 2 | 1.45 | 0.0035 | 1.53 | 0.008 | 0.50/0.41/0.25/ 0.17/0.14 | 0.02/0.01/ 0.01/0.01 | 1.45/1.45/ 1.46/1.47 | 0.0042/0.0039/ 0.0045/0.0047 |
| BB | 0.132 | 4.5 | 0.4 | 0.6 | 4 | 1.52 | 0.025 | 1.53 | 0.008 | 0.50/0.39/0.21/ 0.11/0.08 | 0.06/0.03/ 0.02/0.02 | 1.52/1.51/ 1.52/1.51 | 0.0226/0.0199/ 0.0214/0.0216 |
| DU | 0.1 | 3.4 | 0.6 | 0.8 | 0.066 | 1.53 | 0.008 | 1.53 | 0.008 | 0.50/0.46/0.40/ 0.38/0.37 | 0.09/0.07/ 0.06/0.06 | 1.54/1.51/ 1.52/1.53 | 0.0085/0.0073/ 0.0089/0.0090 |

In the numerical test, the initial guess values (see Section 2.3) are set with the bi-modal combined VPSD and $(n, k)$ of the simulated AERONET product (Table 1), additionally with typical errors of AERONET products (i.e. 0.05 in $n$ and 40% in $k$) (Dubovik et al., 2000), in order to test the scheme tolerance on the initial guess biases. In Fig.3 we present the separated sub-mode CRIs of three typical models and the breaking-down results of VPSD (Fig. 3). It can be seen that both real and imaginary parts of fine and coarse modes are well separated. The maximum error of real part is 0.046 attached to $n_c$ of the biomass burning type, while error of imaginary part is 0.003 for $k_{f,440}$ of the biomass burning. Uncertainty on $n_c$ can be understood as that optical contribution of coarse mode is weak in the case of biomass burning type and thus difficult to be retrieved. Meanwhile, as compensation, the imaginary part $k_f$ is also biased in this case. Moreover, from right column of Fig. 3 we can see that LNM breaking-down are perfectly achieved for each type with very small residuals ($< 6.0 \times 10^{-5}$) which guarantees the retrieval performance on sub-mode CRIs.





**Figure 3.** Estimation of complex refractive indices for both fine (left column) and coarse (middle column) modes of three aerosol types (each row). True values are in circles and retrieved values are in cross symbols. The volume particle size distributions and corresponding breaking-down results (right column) are also shown.

Fig. 4 shows the recovery of $\tau$ and $\tau_a$ (see step e) of Section 2.3) and comparison with true values listed in table 1. In average, we find fairly good agreements for fitting the spectral $\tau$ and $\tau_a$. The absolute $\tau$ error of $1.33 \times 10^{-2}$ at 440 nm in dust type, is relatively larger than other wavelengths but still small enough considering that the AERONET AOD measurement uncertainty is about 0.01-0.02. The maximum error on fitting $\tau_a$ is about $0.55 \times 10^{-2}$ corresponding to biomass burning at 440 nm. In a



meaning of band average, the largest root-mean-square-error (RMSE) of fitting $\tau$ appears in the dust type, corresponds to the underestimate of $n_f$ and $n_c$ (Fig. 3). Similarly, the overestimated $k_{f,440}$ and $k_f$ in the biomass burning type also leads to a relatively larger RMSE ($3.34 \times 10^{-3}$) on fitting $\tau_a$.

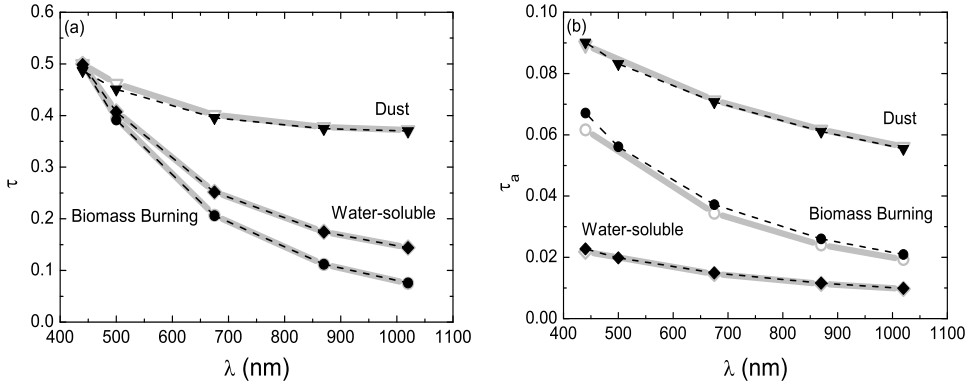

**Figure 4.** Recovery of spectral $\tau$ (left) and $\tau_a$ (right) for three aerosol types (thick line: true, thin line: recovered, corresponding to the separated sub-mode CRIs in Fig.3).

### 3.2 Error estimation

5 In order to evaluate the overall performance of the estimation scheme, we preform numerical experiments to access errors of output sub-mode CRI parameters related to: (i) uncertainties on $\tau$. We set typical $\tau$ uncertainty (here, 0.01) of AERONET products (Holben et al., 1998; Eck et al., 1999) for all $\tau$ bands; (ii) uncertainties on $\tau_a$. Because $\tau_a$ of AERONET is obtained by multiplying $\tau$ with single-scattering albedo ($\omega$), we consider here $\omega$ uncertainty (0.03) of AERONET products (Dubovik et al., 2000) and estimate the uncertainty of $\tau_a$ by error propagation; (iii) uncertainties on VPSD. We set 3 typical VPSD (22 10 bins) uncertainty (15%, 25% and 35% for the average of all bins) of AERONET products following Dubovik et al. (2000) corresponding to the inversion uncertainties on VPSD of water-soluble, biomass burning and dust aerosol type.

Table 2 presents retrieval errors of six sub-mode CRI parameters associated to three typical aerosol types. For the reason of simplification, we only show the impacts of $\Delta\tau = 0.01$, $\Delta\omega = -0.03$ and positive error on VPSD, considering that biases caused by input parameter uncertainties are usually symmetric around real values for such kind of retrieval algorithm (Li et 15 al.. 2006). For the influence of $\Delta\tau = 0.01$, it can be seen that retrieval errors are relatively small. The largest error caused by $\Delta\tau = 0.01$ is on $k_f$ (dust type) which is 0.0023. As to the influence of $\Delta\omega = -0.03$, the retrieval errors increase, especially for the imaginary parts, e.g. $\Delta k_c = 0.0074$ for water-soluble type. The errors caused by uncertainty in VPSD are quite different for real and imagery parts of CRI. The retrieval errors of $n_f$ and $n_c$ are larger than those of $k_f$ and $k_c$, with the maximum $\Delta n_f$ of 0.200 in the case of dust type. As a summary of Table 2, considering that it looks like that not all uncertainties reach 20 the maximum simultaneously, and based on error propagation theory, the total uncertainties on the retrieved sub-mode CRI parameters are estimated as $\Delta n_f = 0.106$, $\Delta k_f = 50.6\%$ ($\Delta k_{f,440} = 75.4\%$); $\Delta n_c = 0.111$, $\Delta k_c = 77.8\%$ ($\Delta k_{c,440} = 56.1\%$), for the average of all typical aerosol types. Or more simplified, the expected errors of sub-mode CRI are $\Delta n_{f/c} = 0.11$ and



**Table 2.** Typical uncertainties on the estimated complex refractive indices of fine ($f$) and coarse ($c$) modes. Three error sources (on $\tau$, $\tau_a$ and VPSD, respectively) and three aerosol types (WS: water-soluble, BB: biomass burning, DU: dust) are considered.

| Error sources | Aerosol types | Fine mode | | | Coarse mode | | |
|---|---|---|---|---|---|---|---|
| | | $\Delta n_f$ | $\Delta k_{f,440}$ | $\Delta k_f$ | $\Delta n_c$ | $\Delta k_{c,440}$ | $\Delta k_c$ |
| $\Delta\tau = 0.01$ | WS | 0.010 | 0.0004 | 0.0003 | 0.006 | 0.0009 | 0.0008 |
| | BB | 0.010 | 0.0001 | 0.0007 | 0.044 | 0.0016 | 0.0014 |
| | DU | 0.026 | 0.0017 | 0.0023 | 0.020 | 0.0003 | 0.0002 |
| $\Delta\omega = -0.03$ (Proxy of $\Delta\tau_a$) | WS | 0.001 | 0.0033 | 0.0001 | 0.037 | 0.0066 | 0.0074 |
| | BB | 0.005 | 0.0073 | 0.0037 | 0.046 | 0.0011 | 0.0012 |
| | DU | 0.008 | 0.0019 | 0.0018 | 0.044 | 0.0021 | 0.0026 |
| $\Delta$VPSD = 15% | WS | 0.048 | 0.0007 | 0.0007 | 0.157 | 0.0032 | 0.0004 |
| 25% | BB | 0.066 | 0.0042 | 0.0045 | 0.060 | 0.0021 | 0.0021 |
| 35% | DU | 0.200 | 0.0072 | 0.0080 | 0.070 | 0.0024 | 0.0080 |
| Total error estimation $^*$ | | 0.106 | 75.41% | 50.62% | 0.111 | 56.05% | 77.76% |

$^*$ Total error estimation $= \sqrt{\overline{x}^2_{\Delta\tau} + \overline{x}^2_{\Delta\tau_a} + \overline{x}^2_{\Delta VPSD}}$ where $\overline{x}$ represents the mean error of sub-CRIs from three aerosol types.

$\Delta k_{f/c(,440)} = 78\%$, which are about 2 times larger than those of AERONET products of all-size CRI (i.e. $\Delta n = 0.04$ and $\Delta k = 40\%$). This is acceptable and logical considering that we are separating mixed information and these uncertainties are still acceptable for most of applications, e.g. validation of chemical models.

### 3.3 Discussion on the sensitivity

As shown in Section 3.1, retrieval performance on the fine and coarse modes can be different with respect to real and imaginary parts of CRI, e.g. for real part, $\Delta n_f$ is significant less than $\Delta n_c$ in the case of biomass burning aerosols. This suggests that the natural properties of aerosol modes may affect the accuracy of sub-mode CRI estimation, and thus it is necessary to perform a simple sensitivity study to further clarify the retrieval possibility and possible limitations, besides the numerical error estimation in Section 3.2.

Firstly, we disturb the scheme outputs (e.g. aerosol sub-CRI parameters) by their expected errors (i.e. $\delta n = 0.111$ and $\delta k = 77.8\%$) as assessed in Section 3.2. Then, by utilizing three aerosol type (WS, BB and DU) models, we trace the effects of these perturbations on the scheme constrain parameters, i.e. $\tau$ and $\tau_a$. Finally, we compare these perturbation results ($\delta\tau/\tau$ and $\delta\tau_a/\tau_a$) with their corresponding sensitivity thresholds (e.g. measurement uncertainties), here $\delta\tau/\tau = 2\%$ and $\delta\tau_a/\tau_a = 6\%$ for AERONET measurements. If the perturbation results are generally beyond the sensitivity thresholds, we can confirm that

the constraint parameters are sensitive to the scheme outputs. It should be mentioned that we employ simultaneously $\tau$ and $\tau_a$ as the constraints in our estimation scheme. Another saying, either sensitivity on $\tau$ or $\tau_a$ will be able to support the estimation of related sub-CRI parameters.





As illustrated in Fig. 5, we find that $\tau$ is mainly sensitive to $n_f$ of the WS and BB types, and their sensitivity curves decrease with the wavelength. Although the relative low sensitivity to $n_f$ presents in DU type, the $\delta\tau/\tau$ is still high than the sensitivity threshold in the case. On the contrary, sensitivity of $k_f$ increases with wavelength, while much higher sensitivity of $k_f$ embodied in $\tau_a$. The sensibilities of $n_c$ of all three types are considerably low, e.g. the largest sensitivity on $\tau_a$ is less than 3%.

5    Meanwhile, $\tau_a$ is sensitive to both $k_c$ and $k_f$ components except for fine mode of dust type and coarse mode of biomass burning type, with the maximum sensitivity of 43% ($k_f$ of biomass burning). The sensitivity of dust type shows a good qualitative agreement with previous studies (Dubovik et al., 2000; Wendisch and von Hoyningen-huene, 1994).

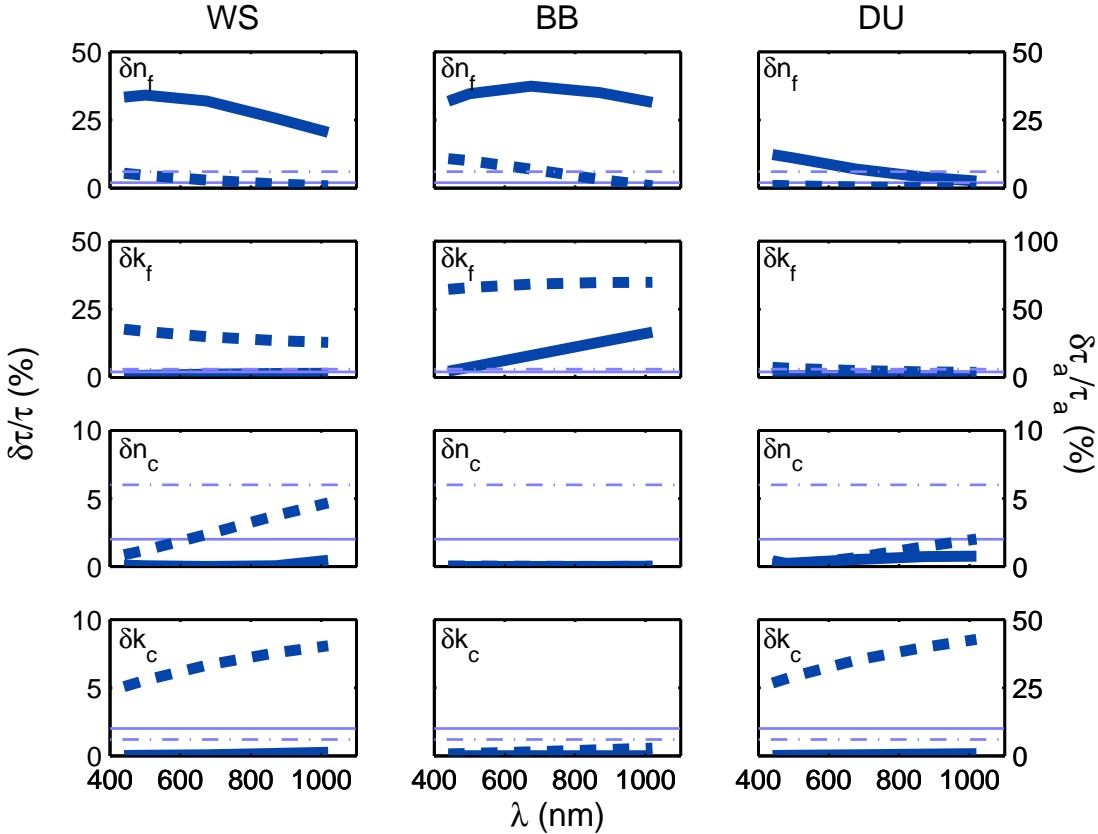

**Figure 5.** Sensitivity study on sub-mode CRI parameters ($n_f$, $k_{f(,440)}$, $n_c$, $k_{c(,440)}$) to constrain parameters ($\tau$ and $\tau_a$) based on three aerosol types (WS: water-soluble, BB: biomass burning, DU: dust). Thick solid line shows the $\delta\tau/\tau$ and thick dash line shows the $\delta\tau_a/\tau_a$. Thin solid and dash line in figure represent the sensitive thresholds (uncertainty of 2% for $\tau$ and 6% for $\tau_a$).

## 4    Test with AERONET Data

The sub-mode CRI estimation scheme can be directly applied to AERONET products. As a realistic test, we chose 1-yr

10    AERONET measurements at Beijing, China, considering its complex aerosol sources and properties resulting from diversified





anthropogenic and natural activities. We utilize AERORNET lev 2.0 (data quality assured) data at Beijing CAMS site from Apr. 2014 to Apr. 2015.

The results show that the mean CRI of fine mode in Beijing is 1.48-0.010i (while imaginary part at 440 nm is 0.012) and that of coarse mode is 1.49-0.004i (while imaginary part at 440 nm is 0.007). These values suggest that fine mode real part refractive

index ($n_f$) are slightly lower than that of coarse mode ($n_c$), while fine mode imaginary part refractive index ($k_f$) are greatly higher than that of coarse mode ($k_c$) in Beijing. Moreover, both fine and coarse modes have larger imaginary parts at 440nm than other wavelengths from 675 to 1020 nm. These results have similar trends as compared with Fig.1 and agree with Hand and Kreidenweis (2002).

For more details, the seasonal mean values of CRIs of fine and coarse modes in Beijing are shown in Fig.6. It can be found

that: (i) both $n_f$ and $n_c$ are the lowest in summer (Fig.6a) which agrees with the maximum humidity of summer in Beijing, because higher aerosol water ($n = 1.33$) content tends to decrease real part of CRI. Meanwhile, we found that the discrepancy between $n_f$ and $n_c$ also reaches the maximum in summer. Considering it is generally recognized that coarse particles are weakly hygroscopic, this discrepancy suggests that hygroscopicity of fine particles are significantly increased in summer under high humidity condition. (ii) For all seasons, $k_c$ is quite constant (Fig.6b). This suggests that large size particulate components

are relatively stable in Beijing. In contrast, $k_f$ shows highly seasonal variation and winter value is about 3 times higher than summer, which can be explained by the increase of carbonaceous component emissions during Beijing's winter heating season (Zhang, Jing et al., 2013). (iii) In Fig.6c, it can be seen that characteristics of ($k_f$ and $k_c$) are similar with that of ($k_{f,440}$ and $k_{c,440}$), except for the enlarged seasonal variation amplitude (especially for $k_{c,440}$). Compared with Fig.1, we thought that this might be caused by Hematite. As jointly seen with Fig.6b, $k_{c,440}$ decreases significantly in summer which may suggest that the

decrease of Hematite is stronger in summer. This indicates some clues on the component changes of coarse particles, e.g. the invaded dust (higher Hematite concentration) might be prohibited significantly in summer due to higher humidity and surface roughness, while the local emission of large particles mainly consist of non-mineral components.

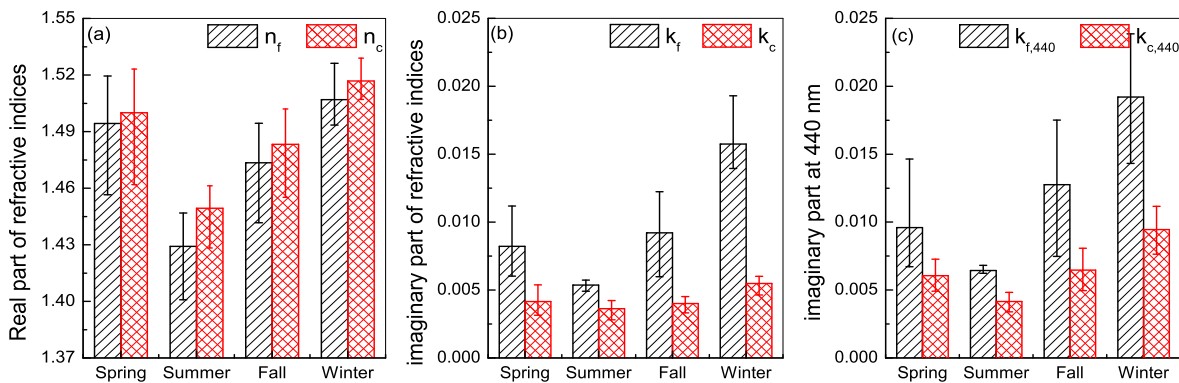

**Figure 6.** Seasonal mean of real part (a), imaginary part (b) and imaginary part at 440nm (c) of sub-mode aerosol refractive indices in Beijing 2014-2015. $f$ and $c$ denote fine and coarse mode respectively. Error bar shows the maximum and minimum of the monthly mean values.





In Fig.7, we illustrate the recovery of scheme input parameters (AERONET $\tau$ and $\tau_a$). It can be seen that the maximum averaged bias (relatively 10% and absolutely 0.029) occurs at 1020 nm. Meanwhile the maximum bias (relatively 11% and absolutely 0.002) in $\tau_a$ is also attached to this longer wavelength. These biases are basically close to our expectation and claimed uncertainties of AERONET products ($\tau$ and $\tau_a$) and thus proves that our sub-mode CRI results are acceptable in the
meaning of optical closure.

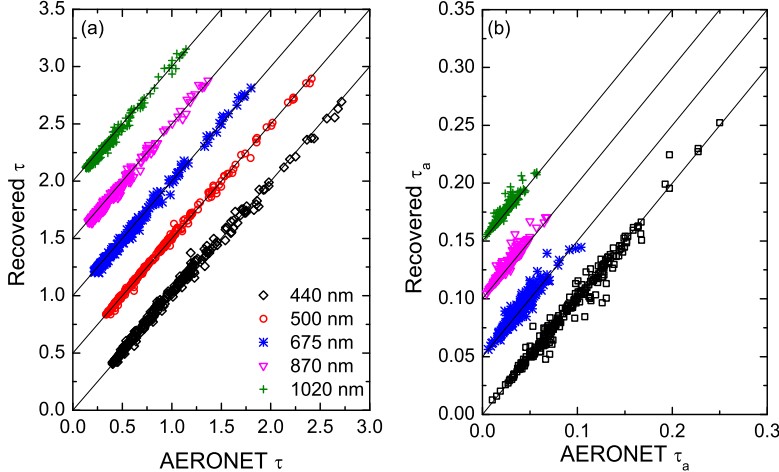

**Figure 7.** Recovery of $\tau$ and $\tau_a$ based on the separated sub-mode aerosol complex refractive indices. N=259 and curves are shifted for a better viewing.

## 5   Conclusions

This paper establishes a scheme to estimate complex refractive indices of both aerosol fine and coarse modes for the total column atmosphere. The input parameters of the scheme are the volume particle size distribution (VPSD), spectral aerosol optical depth ($\tau$) and absorbing aerosol optical depth ($\tau_a$) of AERONET aerosol products, while AERONET complex refractive
indices (CRI) products are used to generate the initial guesses. The retrieval outputs are aerosol CRIs separated for fine ($n_f$, $k_{f,440}$, $k_f$) and coarse modes ($n_c$, $k_{c,440}$, $k_c$) simultaneously. We present the VPSD breaking-down and sub-mode CRI iterative inversion techniques as well as the error estimation and test with AERONET real measurements at Beijing site.

The numerical test with three aerosol types shows that sub-mode CRIs can be well retrieved theoretically with the maximum errors less than about 0.046 (real) and 0.003 (imaginary). The total uncertainties on the retrieved CRIs by considering possible
input AERONET parameter errors together, are about $\Delta n_{f/c} = 0.11$ and $\Delta k_{f/c(,440)} = 78\%$, respectively. Scheme test based on real measurements are performed in AERONET site in Beijing from 2014 to 2015. The results suggest a CRI of 1.48-0.010i ($k_{f,440} = 0.012$) for fine mode particles and 1.49-0.004i ($k_{c,440} = 0.007$) for coarse mode aerosol particles. Retrieval results also reveal that CRIs of both fine and coarse particles have distinct characteristic in summer versus other seasons, which is due to difference of hygroscopic effects on fine and coarse particles, as revealed by separated CRI parameters. Meanwhile, results





suggest that CRIs of fine particles in winter season, especially the imaginary part, are significantly affected by anthropogenic activities, e.g. carbonaceous components from winter heating.

In the next studies, we will focus on the influence of non-sphericity on dust aerosols, which may help to decrease uncertainties on CRIs of coarse mode particles. In addition, this method is not limited to AERONET remote sensing products and also applicable to in situ measurements, e.g. the joint extinction, absorption and size distribution observation obtained from online instruments.

*Acknowledgements.* This work was supported by National Natural Science Fund of China (No. 41601386, 91544219, 41671367) and the Chinese Major Project of High Resolution Earth Observation System (30-Y20A39-9003-15/17).



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
