# Peer review of "Estimation of aerosol complex refractive indices for both fine and coarse modes simultaneously based on AERONET remote sensing products"

_Atmospheric Measurement Techniques, 2017_

## Referee Comment (RC1) · Anonymous Referee #3 · 18 May 2017

The paper presents a method to estimate refractive indices for both fine and coarsemode particles. The retrieval from AERONET assumes size-independent refractive index. The paper assumes that the imaginary part of refractive index has no spectral variation except at 440 nm, while real part of of refractive index has no no spectral variation from 440 - 1020 nm. The size parameters are then derived by fitting lognormal distributions to the inverted size-bin data from AERONET. Mie calculation is conducted to find the sets of both fine and coarse aerosols refractive indices that give the best agreement with AOD and absorbing AOD from AERONET.

Overall, the math in this paper is sound. The results presented for Beijing are interesting. The paper needs more justification of its assumptions and more validation.

1) The assumption that the imaginary part of refractive index has no spectral variation except at 440 nm, while real part of of refractive index has no no spectral variation from 440 - 1020 nm. Is this assumption consistent with the assumption in AERONET's inversion algorithm?

2) In cases/locations when AERONET's inversion shows dominant fraction (~100%) of fine-mode aerosols, its retrieved index of refraction should be considered as appropriate for fine-mode aerosols. The same holds true when aerosols are dominated by coarse particles. There are AERONET sites close to Gobi desert. It will be valuable to look at several cases where dust particles are transported from Gobi desert to Beijing, and compare the retrieved index of refraction for coarse mode in Beijing with that directly retrieved from AERONET site close to the dust source.

---

## Referee Comment (RC2) · Anonymous Referee #4 · 6 Jun 2017

The Authors present a method for the separate estimation of the aerosol refractive index from AERONET data. First they fit AERONET aerosol size distribution to a multimodal log-normal distribution, then they group the modes of the fitted log-normal distribution into a "fine" and a "coarse" mode, and then they proceed to the estimate of the refractive index of each mode by an iterative fitting AERONET total and absorption AOTs to Mie forward calculations. The proposed method looks fine to me. The steps of the procedure are well identified, the underlying assumptions are clearly stated and so are the limitations of the method (e.g., not taking the possibility of nonspherical particles into account). The validation on synthetic data, instead, looks a bit shallow, because the Authors only test three configurations, in which three realistic fine mode

aerosol types (water-soluble, biomass burning and dust) are combined with a "default" coarse mode with refractive index 1.53+i0.008: in this section I would have been curious to see tests with more combinations of aerosol parameters. Anyway, in the last section of the paper the Authors also make the effort of applying their method to real AERONET measurements taken at Beijing, and they show that their separate retrieval allows a reasonable physical interpretation (which is probably the best possible "validation" of a refractive index retrieval, given that independent correlative measurements of this parameter are very difficult to obtain. and even an objective definition of the refractive index of a mixture of aerosol components is problematic in itself). Furthermore, the Authors show that their multicomponent refractive index retrievals fit AERONET AOTs quite well. In view of this, I think this paper can be published with minor revisions. I would recommend, though, a proofreading by a native English speaker, because the quality of the written English looks below par in some parts of the manuscript. Below are some suggestions for the modification of some unclear statements, and some other minor comments.

**MINOR COMMENTS**

- P1, L4-5. I would suggest to change "... based on AERONET aerosol products, including" etc., with "... based on AERONET volume particle size distribution" etc.
- P1, L5-6. The sentence "The method ... simultaneously" is a bit unclear. Consider removing it or rephrasing with something like "The method consists of two steps. First a multimodal log-normal distribution that best approximates the AERONET VPSD is found. Then the fine and coarse mode CRIs are found by iterative fitting of AERONET AODs to Mie calculations."
- P2, L8-9. I do not understand what the last two sentences mean. Especially the last one ("Raul and Chazette etc.").

- P2, L17. Change "There are only few studies ... attempted..." to "Only a few studies ... attempted...". Furthermore, Wu et al. (2015) does not describe re-trieval from ground-based measurements. It describes retrievals from airborne measurements.
- P3, L12.  $\sigma_i$  is the "geometric standard deviation" (not the ordinary one) of r for each mode. As an alternative, if you prefer not introducing the concept of geometric standard deviation, you can say that " $\ln \sigma_i$  is the standard deviation of  $\ln r$  for each mode".
- P5, L1. Cite:

J. A. Nelder, and R. Mead (1965), "A simplex method for function minimization". Comp. J., 7, 308-313, doi: 10.1093/comjnl/7.4.308

- P9, L1. What does "In a meaning of band average" mean?
- P10, L16-17. I do not understand the meaning of the last sentence: "Either sensitivity on  $\tau$  or  $\tau_a$  will be able to support the estimation of related sub-CRI parameters".

**TECHNICAL CORRECTIONS**

- P2, L16. Consider moving "inventories" before the parenthesis.
- P2, L17. P2, L17. "knowledge of ... are essential" -> "... is essential ... "
- P3, L21. "subsequence" -> "subsequent"
- P5, L26, "an" -> "a"
- P5, L28. "achieves" -> is achieved. "If yes" -> "If so"

**СЗ**

- P9, L5. "preform" -> "perform". "access" -> "assess" ?
- P9, L18. "imagery" -> "imaginary"
- P10, L16. "Another saying" -> "In other words"
- P11, L2-3. "relative" -> "relatively", "presents" -> "present", "high" -> "higher", "in the case" -> "for this case" ?
- P11, L4. "sensibilities" -> "sensitivities"
- P12, L13. "...the hygroscopicity ... are significantly increased" -> "... is significantly increased"
- P12, L15-16. Consider removing the parentheses from " $(k_f \text{ and } k_c)$ ", and from " $(k_{f,440} \text{ and } k_{c,440})$ ".
- P14, L5. What does "online" mean in this sentence?

---

## Author Response (AR1)

Dear reviewer:

Thank you very much for your careful review and constructive suggestions with regard to this manuscript. We appreciate for Reviewer's work earnestly, and hope that the corrections will meet with approval. Please find below our detailed responses to reviewer's question and comments.

**Referee #3**
**The paper presents a method to estimate refractive indices for both fine and coarse mode particles. The retrieval from AERONET assumes size-independent refractive index. The paper assumes that the imaginary part of refractive index has no spectral variation except at 440 nm, while real part of of refractive index has no no spectral variation from 440 - 1020 nm. The size parameters are then derived by fitting lognormal distributions to the inverted size-bin data from AERONET. Mie calculation is conducted to find the sets of both fine and coarse aerosols refractive indices that give the best agreement with AOD and absorbing AOD from AERONET. Overall, the math in this paper is sound. The results presented for Beijing are interesting. The paper needs more justification of its assumptions and more validation.**

**1) The assumption that the imaginary part of refractive index has no spectral variation except at 440 nm, while real part of of refractive index has no no spectral variation from 440 - 1020 nm. Is this assumption consistent with the assumption in AERONET's inversion algorithm?**

**Response**: Firstly, this assumption is only set to output (CRI of separated fine & coarse modes) of this work, instead of inputs (i.e. the AERONET's inversion products still keep their spectral variation). The objective of this work (separating CRI of fine & coarse modes) focuses on improving the inference of aerosol component information. Figure 1 shows the real parts (n) of the majority of aerosol components are quite constant from UV to near infrared spectral region, and imaginary parts show a significant spectral variation at short wavelengths (e.g. 440nm). This explains our basic consideration on the assumption of the spectral behaviors of output CRIs. In addition, the AERONET algorithm paper stated the similar consideration (Dubovik & King, JGR, 2000):

"*Spectral variability is usually **not expected for both real and imaginary parts** of the aerosol particle refractive index. For example, the widely cited paper by Shettle and Fenn [1979] shows practically wavelength-independent complex refractive indices in the spectral interval of interest (440-1020 nm) for the materials typically composing atmospheric aerosols. Similarly, aerosol models by Tanre et al. [1999] assume single constant values of complex refractive index for the spectral interval considered.*"

Secondly, we think that this assumption is not in confliction with AERONET's

algorithm. The AERONET inversion algorithm assumes identical real and imaginary parts of the refractive indices for both fine and coarse modes coincidently, but allowing independent values at each wavelength (440, 675, 870 and 1020nm). This is mainly to deal with the mixture of mode/components, which can be seen (note: there, "aerosol particles" means mixture of fine & coarse modes) in their paper (Dubovik & King, JGR, 2000):

*"However, in the scientific literature there are multiple indications of possible spectral selectivity of the refractive index for aerosol particles [e.g., Ackerman and Toon, 1981; Patterson and McMahon, 1984; Horvath, 1993; Dubovik et al., 1998b, Yumasoe et al., 1998]. Therefore we constrain the spectral variability of the retrieved complex refractive index to some practically reasonable ranges rather than to any particular model of the atmospheric aerosol."*

**2) In cases/locations when AERONET's inversion shows dominant fraction (~100%) of fine-mode aerosols, its retrieved index of refraction should be considered as appropriate for fine-mode aerosols. The same holds true when aerosols are dominated by coarse particles. There are AERONET sites close to Gobi desert. It will be valuable to look at several cases where dust particles are transported from Gobi desert to Beijing, and compare the retrieved index of refraction for coarse mode in Beijing with that directly retrieved from AERONET site close to the dust source.**

**Response**: According to the Reviewer's comments, we choose a dust event period from Apr. 17-19 2017 both at Beijing site and Dalanzadgad site close to Gobi desert in Mongolia (Fig. S1a). The dust aerosol in Beijing transported from Dalanzadgad site can be seen clearly in Fig 1b, simulated by HYSPLIT model reached Apr. 20, 2017. The high concentrations of volume particle size distributions in coarse mode (Fig. S2) are similar at Dalanzadgad and Beijing site. It is indicate that the similar properties of dust can be observed at both Beijing and Dalanzadgad sites. Fig. S3(a) and (b) shows a fairly good consistency of the real parts (n) at Dalanzadgad from AERONET and the retrievals for coarse mode in Beijing from our algorithm and close to 1.6, although the observed time is not an exact match. But the imaginary parts do not agree each other. It can be explained by the variation of transported aerosol properties. In addition, some uncertainties can be involved in analysis and retrieval because the AERONET Lev 1.5 data is used in this part. Particularly, the strong absorption (large $k$) in Dalanzadgad site on Apr. 17 is inveracious.

[Figure]

Fig. S1 (a) Map of site locations; (b) backward trajectory on Apr. 17-19, 2017

[Figure]

Fig. S2 volume particle size distributions of (a) Dalanzadgad and (b) Beijing site during Apr. 17-19, 2017.

[Figure]

[Figure]

Fig. S3 The comparison of CRI from AERONET at Dalanzadgad site (a, c) and retrieved one in coarse mode at Beijing site (b, d).

To further detect the accuracy of coarse mode, additional three typical dust aerosol model (Dubovik et al., 2002) are employed and combined with the fine-mode WS and BB (Table 1). As the Table S1 shows, The error of real part in coarse mode is lower than 0.1, and the retrieved imaginary part of CRI in coarse mode is more accurate with the error of less than 0.003 except for the biomass burning model.

Table S1. The retrieved errors of typical dust aerosol models.

| Aerosol model | Input CRI in coarse mode | | | Error in coarse mode | | |
|---|---|---|---|---|---|---|
| | $n_c$ | $k_{c,440}$ | $k_c$ | $n_c$ | $k_{c,440}$ | $k_c$ |
| WS | 1.55 | 0.0025 | 0.001 | -0.091 | 0.000 | 0.003 |
| | 1.56 | 0.0029 | 0.001 | -0.099 | 0.000 | 0.003 |
| | 1.48 | 0.0025 | 0.0006 | -0.033 | 0.000 | 0.003 |
| BB | 1.55 | 0.0025 | 0.001 | -0.013 | 0.018 | 0.022 |
| | 1.56 | 0.0029 | 0.001 | -0.022 | 0.018 | 0.021 |
| | 1.48 | 0.0025 | 0.0006 | 0.054 | 0.019 | 0.022 |
| DU | 1.55 | 0.0025 | 0.001 | -0.008 | 0.000 | 0.000 |
| | 1.56 | 0.0029 | 0.001 | -0.009 | 0.000 | 0.000 |
| | 1.48 | 0.0025 | 0.0006 | -0.010 | 0.000 | 0.000 |

Dubovik O., Holben B., Eck T. F., Smirnov A., Kaufman Y. J., King M. D., Tanre D. and Slutsker I. (2002), Variability of absorption and optical properties of key aerosol types observed in worldwide locations, Journal of the atmospheric sciences, 59, 590-608.

**Referee #4**
**The Authors present a method for the separate estimation of the aerosol refractive index from AERONET data. First they fit AERONET aerosol size distribution to a multimodal log-normal distribution, then they group the modes of the fitted log-normal distribution into a "fine" and a "coarse" mode, and then**

they proceed to the estimate of the refractive index of each mode by an iterative fitting AERONET total and absorption AOTs to Mie forward calculations. The proposed method looks fine to me. The steps of the procedure are well identified, the underlying assumptions are clearly stated and so are the limitations of the method (e.g., not taking the possibility of nonspherical particles into account). The validation on synthetic data, instead, looks a bit shallow, because the Authors only test three configurations, in which three realistic fine mode aerosol types (water-soluble, biomass burning and dust) are combined with a "default" coarse mode with refractive index $1.53+i0.008$: in this section I would have been curious to see tests with more combinations of aerosol parameters. Anyway, in the last section of the paper the Authors also make the effort of applying their method to real AERONET measurements taken at Beijing, and they show that their separate retrieval allows a reasonable physical interpretation (which is probably the best possible "validation" of a refractive index retrieval, given that independent correlative measurements of this parameter are very difficult to obtain. and even an objective definition of the refractive index of a mixture of aerosol components is problematic in itself). Furthermore, the Authors show that their multicomponent refractive index retrievals fit AERONET AOTs quite well. In view of this, I think this paper can be published with minor revisions. I would recommend, though, a proofreading by a native English speaker, because the quality of the written English looks below par in some parts of the manuscript. Below are some suggestions for the modification of some unclear statements, and some other minor comments.

**MINOR COMMENTS**

**(1) P1, L4-5. I would suggest to change "…based on AERONET aerosol products, including" etc., with "…based on AERONET volume particle size distribution" etc.**

Response: We have corrected according to the Reviewer's comments.
"This paper establishes a method to separate CRIs of fine and coarse particles based on AERONET volume particle size distribution (VPSD), aerosol optical depth (AOD) and absorbing AOD."

**(2) P1, L5-6. The sentence "The method … simultaneously" is a bit unclear. Consider removing it or rephrasing with something like "The method consists of two steps. First a multimodal log-normal distribution that best approximates the AERONET VPSD is found. Then the fine and coarse mode CRIs are found by iterative fitting of AERONET AODs to Mie calculations."**

Response: We have corrected according to the Reviewer's comments.
"The method consists of two steps. First a multimodal log-normal distribution that

best approximates the AERONET VPSD is found. Then the fine and coarse mode CRIs are found by iterative fitting of AERONET AODs to Mie calculations."

**(3) P2, L8-9. I do not understand what the last two sentences mean. Especially the last one ("Raul and Chazette etc.").**

Response: We have corrected the sentences in the manuscript.
"In addition, Li et al. (2006) further added the polarized sky radiance measurements to the inversion algorithm in order to better constrain AERONET CRI retrievals. The Lidar measurements are also used to obtain CRI of aerosols within planetary boundary layer (Raut and Chazette, 2007)."

**(4) P2, L17. Change "There are only few studies … attempted..." to "Only a few studies … attempted...". Furthermore, Wu et al. (2015) does not describe retrieval from ground-based measurements. It describes retrievals from airborne measurements.**

Response: We have corrected according to the Reviewer's comments.
"Only few studies (e.g. Xu et al., 2015; Wu et al., 2015) attempted to retrieve CRI of both fine and coarse modes simultaneously from advanced remote sensing measurements, e.g. multi-spectral polarized sky radiance."

**(5) P3, L12. $\sigma_i$ is the "geometric standard deviation" (not the ordinary one) of r for each mode. As an alternative, if you prefer not introducing the concept of geometric standard deviation, you can say that "$\ln\sigma_i$ is the standard deviation of ln r for each mode".**

Response: We have corrected according to the Reviewer's comments.
"$C_i$ ($\mu m^3/\mu m^2$) and $r_i$ ($\mu m$) and $\ln \sigma_i$ are the volume modal concentration, median radius and standard deviation of $\ln r_i$ for each LNM mode, respectively."

**(6) P5, L1. Cite:**
**J. A. Nelder, and R. Mead (1965), "A simplex method for function minimization". Comp. J., 7, 308-313, doi: 10.1093/comjnl/7.4.308**

Response: We have cited the paper according to the Reviewer's comments.
"…, an iterative procedure is performed by minimizing Chi-Square on VPSD (see Eq.5) using the NelderMead simplex algorithm (Nelder and Mead, 1965; Lagarias et al., 1998)."

**(7) P9, L1. What does "In a meaning of band average" mean?**

Response: "In a meaning of band average" means the RMSE is calculated in each wavelength and the largest RMSE in five wavelength appears in the dust type. We

rewrite the sentence as follows:

"The largest root-mean-square-error (RMSE) of fitting $\tau$ appears in the dust type, corResponse to the underestimate of $n_f$ and $n_c$ (Fig. 3)."

**(8) P10, L16-17. I do not understand the meaning of the last sentence: "Either sensitivity on $\tau$ or $\tau_a$ will be able to support the estimation of related sub-CRI parameters".**

Response: This is an English expression problem. We replaced the sentence by "this suggest that both $\tau$ and $\tau_a$ sensitivities contribute to the convergence of the iterative scheme. Given only one information ($\tau$ or $\tau_a$) is sensitive, it is still possible to constrain the scheme give its sensitivity is strong enough."

**TECHNICAL CORRECTIONS**

**(1) P2, L16. Consider moving "inventories" before the parenthesis.**

Response: We have corrected according to the Reviewer's comments.
"…, aerosols radiative properties are simulated based on source emission inventories (i.e. fine and coarse sources separately), …"

**(2) P2, L17. P2, L17. "knowledge of . . . are essential" -> ". . . is essential . . . "**

Response: We have corrected according to the Reviewer's comments.
"…, and thus knowledge on CRI of different aerosol modes is essential to validate model performance for the assessment of aerosol climate effects."

**(3) P3, L21. "subsequence" -> "subsequent"**

Response: We have corrected according to the Reviewer's comments.
"The above assumed spectral properties of sub-CRIs are useful to simplify subsequent procedure and it basically fits current knowledge on aerosol properties."

**(4) P5, L26, "an" -> "a"**

Response: We have corrected according to the Reviewer's comments.
"e). Find the optimal solution based on a Limited-memory optimization algorithm (BFGS: Broyden–Fletcher–Goldfarb–Shanno) (Zhu et al., 1997) by constraining both $\tau(\lambda)$ and $\tau_a(\lambda)$ with AERONET products."

**(5) P5, L28. "achieves" -> is achieved. "If yes" -> "If so"**

Response: We have corrected according to the Reviewer's comments.
"Check if the convergence, $(f_i - f_{i+1})/\max(f_{i+1}, f_i, 1) < \eta \times \varepsilon$, is achieved. If so, output the separated sub-CRI parameters, …"

**(6) P9, L5. "preform" -> "perform". "access" -> "assess" ?**

Response: We have corrected according to the Reviewer's comments.
"In order to evaluate the overall performance of the estimation scheme, we perform numerical experiments to assess errors of output sub-mode CRI parameters related to: …"

**(7) P9, L18. "imagery" -> "imaginary"**

Response: We have corrected according to the Reviewer's comments.
"The errors caused by uncertainty in VPSD are quite different for real and imaginary parts of CRI."

**(8) P10, L16. "Another saying" -> "In other words"**

Response: We have corrected according to the Reviewer's comments.
"In other words, this suggest that both $\tau$ and $\tau_a$ sensitivities contribute to the convergence of the iterative scheme. Given only one information ($\tau$ or $\tau_a$) is sensitive, it is still possible to constrain the scheme give its sensitivity is strong enough."

**(9) P11, L2-3. "relative" -> "relatively", "presents" -> "present", "high" -> "higher", "in the case" -> "for this case" ?**

Response: We have corrected according to the Reviewer's comments.
"Although the relatively low sensitivity to $n_f$ present in DU type, the $\delta\tau/\tau$ is still higher than the sensitivity threshold for this case."

**(10) P11, L4. "sensibilities" -> "sensitivities"**

Response: We have corrected according to the Reviewer's comments.
"The sensitivities of $n_c$ of all three types are considerably low, …"

**(11) P12, L13. ". . . the hygroscopicity . . . are significantly increased" -> ". . . is significantly increased"**

Response: We have corrected according to the Reviewer's comments.
"…, this discrepancy suggests that hygroscopicity of fine particles is significantly increased in summer under high humidity condition."

**(12) P12, L15-16. Consider removing the parentheses from "(kf and kc)", and**

**from "(kf;440 and kc;440)".**

Response: We have corrected according to the Reviewer's comments.
"(iii) In Fig.6c, it can be seen that characteristics of $k_f$ and $k_c$ are similar with that of $k_{f,440}$ and $k_{c,440}$, except for the enlarged seasonal variation amplitude (especially for $k_{c,440}$)."

**(13) P14, L5. What does "online" mean in this sentence?**

Response: The "online" means "real-time". We rewrite this sentence as follows:
"…, e.g. the joint extinction, absorption and size distribution observation obtained from measurements in real-time."

[revised manuscript text omitted]
(r)\right)\right), \ i = 1, m \\ \sigma_i = \sqrt{r_i^+ / r_i^-} \\ C_i = v(r_i) \end{cases} \tag{4}$$

where, $v$ (instead of d$V$/dln$r$) is AERONET VPSD, and $r_i^+$ and $r_i^-$ are the zero-crossing points of $g(r)$ around $r_i$ in each mode. Then, to obtain the optimized values of these three parameters (i.e. $C_i$, $r_i$ and $\sigma_i$) of each peak, an iterative procedure is performed by minimizing Chi-Square on VPSD (see Eq.5) using the NelderMead simplex algorithm (Nelder and Mead, 1965; Lagarias et al., 1998).

$$\chi^2 = \sum_{j=1}^{22} \frac{\left(v(r_j) - v_{calc}(r_j)\right)^2}{v_{(r_j)}}, \quad j = 1, 22 \tag{5}$$

where, $v$ and $v_{calc}$ are d$V$/dln$r$ from AERONET products and re-calculated from the separated fine and coarse-mode parameters, respectively. And $j$ is the bins of AERONET VPSD.

**2.3 Separating refractive indices for fine and coarse modes**

According to the aerosol characterization framework in Section 2.1, the flowchart (Fig. 2) of the fine and coarse mode CRI separation is as follows (based on the separated size distribution in Section 2.2):

a). Guesses of 6 output parameters ($n_f$, $k_{f,440}$, $k_c$; $n_c$, $k_{c,440}$, $k_c$). The initial guess values are set with AERONET product values: $n_f = n_{AERONET}(440 \text{ nm})$, $k_{f,440}$ and $k_f = k_{AERONET}(440 \text{ nm})$, $n_c = n_{AERONET}(870$ nm), $k_{c,440}$ and $k_c = k_{AERONET}(870 \text{ nm})$. Meanwhile, the boundary ranges of these parameters are set as: $n_f$ [1.33, 1.6], $n_c$ [1.33, 1.6], $k_{f,440}$ [0.0, 0.5], $k_{c,440}$ [0.0, 0.5], $k_f$ [0.0001, 0.5] and $k_c$ [0.0001, 0.5].

b). Calculating effective CRI corresponding to each VPSD bins. Here, based on the guessed CRIs of both fine and coarse modes in the previous step, we employ an internal mixing approach, following volume average rule (Heller, 1965), to estimate CRI of each particle radius bins:

$$\begin{aligned} n(r) &= \frac{n_f v_f(r) + n_c v_c(r)}{v_f(r) + v_c(r)} \\ k(\lambda, r) &= \frac{k_f v_f(r) + k_c v_c(r)}{v_f(r) + v_c(r)} \end{aligned} \
[revised manuscript text omitted]
. In other words, this suggest that both $\tau$ and $\tau_a$ sensitivities contribute to the convergence of the iterative scheme. Given only one information ($\tau$ or $\tau_a$) is sensitive, it is still possible to constrain the scheme give its sensitivity is strong enough. Another saying, either sensitivity on $\tau$ or $\tau_a$ will be able to support the estimation of related sub-CRI parameters.

As illustrated in Fig. 5, we find that $\tau$ is mainly sensitive to $n_f$ of the WS and BB types, and their sensitivity curves decrease with the wavelength. Although the relatively low sensitivity to $n_f$ presents in DU type, the $\delta\tau/\tau$ is still higher than the sensitivity threshold for this case. On the contrary,

sensitivity of $k_f$ increases with wavelength, while much higher sensitivity of $k_f$ embodied in $\tau_a$. The  sensitivities of $n_c$ of all three types are considerably low, e.g. the largest sensitivity on $\tau_a$ is less than 3%. Meanwhile, $\tau_a$ is sensitive to both $k_c$ and $k_f$ components except for fine mode of dust type and coarse mode of biomass burning type, with the maximum sensitivity of 43% ($k_f$ of biomass burning). The sensitivity of dust type shows a good qualitative agreement with previous studies (Dubovik et al., 2000; Wendisch and von Hoyningen-huene, 1994).

[Figure]

**Fig. 5** Sensitivity study on sub-mode CRI parameters ($n_f$, $k_{f(,440)}$, $n_c$, $k_{c(,440)}$) to constrain parameters ($\tau$ and $\tau_a$) based on three aerosol types (WS: water-soluble, BB: biomass burning, DU: Dust). Thick solid line shows the $\delta\tau/\tau$ and thick dash line shows the $\delta\tau_a/\tau_a$. Thin solid and dash line in figure represent the sensitive thresholds (uncertainty of 2% for $\tau$ and 6% for $\tau_a$).

It should be noted that we employ together $\tau$ and $\tau_a$ and equally weight them in the kernel function of our estimation scheme. Another saying, either sensitivity on $\tau$ or $\tau_a$ will be able to support the successful estimation of related sub-CRI parameters. Only in the cases that both $\tau$ and $\tau_a$ have no sensitivity, we will have difficulties in the retrieval. While more accurate initial guess values are expected to decrease the uncertainty, e.g. on $n_c$ suggested by Xu et al. (2015).

[revised manuscript text omitted]

Nelder J. A. and Mead R. (1965), A simplex method for function minimization, Comp. J., 7, 308-313, doi: 10.1093/comjnl/7.4.308.

Nilsson B. (1979), Meteorological influence on aerosol extinction in the 0.2–40-mm wavelength range, Appl. Opt., 18(20), 3457– 3473.

Palmer K. F., and Williams D. (1975), Optical constants of sulfuric acid -- Application to clouds of Venus, Appl. Opt., 14(1), 208– 219.

Patterson E. M., Gillete D. A., and Stockton B. H. (1977), Complex index of refraction between 300 and 700 nm for Saharan aerosol, J. Geophys. Res., 82, 3153–3160.

Raut J.-C. and Chazette P. (2007), Retrieval of aerosol complex refractive index from a synergy between lidar, sunphotometer and in situ measurements during LISAIR experiment, Atmos. Chem. Phys., 7, 2797–2815.

Schuster, G. L., Dubovik, O., and Arola, A. (2015), Remote sensing of soot carbon – Part 1: Distinguishing different absorbing aerosol species, Atmos. Chem. Phys. Discuss., 15, 13607-13656, doi:10.5194/acpd-15-13607-2015.

Senftleben H., and Benedict E. (1917), Über die optischen Konstanten und die Strahlungsgesetze der Kohle, Annalen der Physik, 54, 65–78.

Sinyuk A., Torres O., and Dubovik O. (2003), Combined use of satellite and surface observations to infer the imaginary part of refractive index of Saharan dust, Geophys. Res. Lett., 30(2), 1081, doi:10.1029/2002GL016189.

Sokolik I. N., and Toon O. B. (1999), Incorporation of mineralogical composition into models of the radiative properties of mineral aerosol from UV to IR wavelengths, J. Geophys. Res., 104(D8), 9423– 9444.

Volz, F. E. (1973), Infrared optical constants of ammonium sulfate, Sahara dust, volcanic pumice,

and fly ash, Appl. Opt., 12, 564-568.

Wagner R., Ajtai T., Kandler K., Lieke K., Linke C., Muller T., Schnaiter M., and Vragel M. (2012), Complex refractive indices of Saharan dust samples at visible and near UV wavelengths: a laboratory study, Atmos. Chem. Phys., 12, 2491-2512.

Wendisch M. and von Hoyningen-Huene W. (1994), Possibility of refractvie index determination of atmospheric aerosol particles by ground-based solar extinction and scattering measurements, Atmospheric Environment, 28(5), 785-792.

Willeke K., Whitby K. T. (1975), Atmospheric aerosols: size distribution interpretation. J. Air. Pollut. Contr. Assoc., 25(5), 529–534.

Woo C., You S., and Lee J. (2013), Determination of refractive index for absorbing spheres, Optik, 124, 5254– 5258.

Wu L.,Hasekamp O., van Diedenhoven B. and Cairns B. (2015), Aerosol retrieval from multiangle, multispectral photopolarimetric measurements: importance of spectral range and angular resolution, Atmos. Meas. Tech., 8, 2625-2638.

Xu, X., Wang J., Zeng J., Spurr R., Liu X., Dubovik O., Li L., Li Z., Mishchenko M. I., Siniuk A., and Holben B.N. (2015), Retrieval of aerosol microphysical properties from AERONET photo-polarimetric measurements: 2. A new research algorithm and case demonstration, J. Geophys. Res. Atmos., 120(14), 7079-7098, doi:10.1002/2015JD023113.

Zhang R., Jing J., Tao J., Hsu S.-C., Wang G., Cao J., Lee C. S. L., Zhu L., Chen Z., Zhao Y., and Shen Z. (2013), Chemical characterization and source apportionment of $PM_{2.5}$ in Beijing: seasonal perspective, Atmos. Chem. Phys., 13, 7053–7074.

Zhang Y., Li Z., Wang Y., Li K., Li D., Zhang Y. H., Wei P., Wang L., Lv Y. (2013), Improving accumulation mode fraction based on spectral aerosol optical depth in Beijing. Spectroscopy and Spectral Analysis, 33, 2795–2802.

Zhu C., Byrd R. H. and Nocedal J. (1997), L-BFGS-B: Algorithm 778: L-BFGS-B, FORTRAN routines for large scale bound constrained optimization, ACM Transactions on Mathematical Software, 23(4), pp. 550 − 560.